# Transcriptional Changes in Potato Sprouts upon Interaction with *Rhizoctonia solani* Indicate Pathogen-Induced Interference in the Defence Pathways of Potato

**DOI:** 10.3390/ijms22063094

**Published:** 2021-03-18

**Authors:** Rita Zrenner, Bart Verwaaijen, Franziska Genzel, Burkhardt Flemer, Rita Grosch

**Affiliations:** 1Leibniz Institute of Vegetable and Ornamental Crops (IGZ), Plant-Microbe Systems, Theodor-Echtermeyer-Weg 1, 14979 Großbeeren, Germany; bverwaai@cebitec.uni-bielefeld.de (B.V.); f.genzel@fz-juelich.de (F.G.); b.flemer@ikmb.uni-kiel.de (B.F.); grosch@igzev.de (R.G.); 2Faculty of Biology/Computational Biology, Bielefeld University, 26 Universitätsstr. 27, 33615 Bielefeld, Germany; 3Institute of Bio- and Geosciences IBG-2, Plant Sciences, Forschungszentrum Jülich GmbH, 52425 Jülich, Germany; 4Institute of Clinical Molecular Biology (IKMB), Kiel University, Rosalind-Franklin-Straße 12, 24105 Kiel, Germany

**Keywords:** *Solanum tuberosum*, plant disease, black scurf disease, RNA-sequencing, *Rhizoctonia solani*

## Abstract

*Rhizoctonia solani* is the causer of black scurf disease on potatoes and is responsible for high economical losses in global agriculture. In order to increase the limited knowledge of the plants’ molecular response to this pathogen, we inoculated potatoes with *R. solani* AG3-PT isolate Ben3 and carried out RNA sequencing with total RNA extracted from potato sprouts at three and eight days post inoculation (dpi). In this dual RNA-sequencing experiment, the necrotrophic lifestyle of *R. solani* AG3-PT during early phases of interaction with its host has already been characterised. Here the potato plants’ comprehensive transcriptional response to inoculation with *R. solani* AG3 was evaluated for the first time based on significantly different expressed plant genes extracted with DESeq analysis. Overall, 1640 genes were differentially expressed, comparing control (−Rs) and with *R. solani* AG3-PT isolate Ben3 inoculated plants (+Rs). Genes involved in the production of anti-fungal proteins and secondary metabolites with antifungal properties were significantly up regulated upon inoculation with *R. solani*. Gene ontology (GO) terms involved in the regulation of hormone levels (i.e., ethylene (ET) and jasmonic acid (JA) at 3 dpi and salicylic acid (SA) and JA response pathways at 8 dpi) were significantly enriched. Contrastingly, the GO term “response to abiotic stimulus” was down regulated at both time points analysed. These results may support future breeding efforts toward the development of cultivars with higher resistance level to black scurf disease or the development of new control strategies.

## 1. Introduction

Potato (*Solanum tuberosum* L.) is the fourth most important food crop after corn, rice and wheat. Global harvest in 2018 amounted to 368 tons of fresh tubers from 17.6 million ha, with more than 80% of produce coming from Asia and Europe (http://www.fao.org/faostat/en/#home, accessed on 10 March 2021). The potato yield per unit area varied depending on the production site from less than 5000 kg/ha in some African countries to 50,000 kg/ha in Europe and New Zealand, with a worldwide average of 20,944 kg/ha (http://www.fao.org/faostat/en/#home, accessed on 10 March 2021). Pathogens are estimated to reduce crop yield by 14% globally and approximately 75% of total potato yield may be lost to pests if no crop protection strategies would be in place [1]. Besides the oomycete pathogen *Phytophthora infestans*, the necrotrophic pathogen *Rhizoctonia solani* [teleomorph *Thanatephorus cucumeris* (A. B. Frank) Donk] is one of the major fungal potato pathogens with global distribution [2]. The dominant anastomosis group (AG) on potato is AG3 [3]. The most visible symptoms are commonly called black scurf, where sclerotia form on tubers, and sunken brown necrotic lesions on stems, stolons and roots are further indications of infections with *R. solani* [4]. The control of *R. solani* in potato is difficult and based mainly on the use of chemical fungicides. However, the current control methods are not efficient enough to avoid losses by black scurf disease and the problem seems to increase in practice. The use of resistant cultivars represents an efficient alternative control strategy, but no fully resistant cultivars are available. While quantitative differences in the degree of susceptibility to black scurf disease exist between cultivars based on field observations [5], the molecular response of potato due to *R. solani* attack is largely unknown.

Perception of a pathogen by the plant triggers a hierarchy of regulatory genes which finally results in biosynthesis of metabolites and proteins able to suppress pathogens directly [6]. Although previous works reported that mainly the hormone jasmonic acid (JA) is primarily induced by necrotrophic pathogens, it was shown that genes associated with salicylic acid (SA) as well as JA/ethylene (ET) pathways were up regulated in root and shoot tissue of potato in response to *R. solani* infection [7,8]. However, these studies were based on analysis of expression level of selected common defence-related genes [7] or on microarray analysis [8]. In contrast, RNA sequencing (RNA-seq) is a promising tool that does not rely on any prior knowledge of transcripts and allows for the study of host–pathogen interaction in the same tissue concomitantly [9]. It has already been successfully used to investigate the early transcriptomic reaction of *R. solani* during interaction with its host potato and revealed a necrotrophic lifestyle of the fungus in this initial phase [10]. Hence, this technology is a powerful tool to simultaneously identify transcripts of potato in response to the soil-borne pathogen *R. solani* AG3-PT in the identical experiment with a dual RNA-seq approach. A comprehensive understanding of host response due to pathogen attack requires knowledge of changes in the expression of associated genes during the course of interaction.

Here, we report the first transcriptomes at two different time points of potato in response to compatible interaction with the necrotrophic pathogen *R. solani* AG3-PT. To understand defence-associated processes and the systemic host response at molecular level can support the discovery of novel mechanisms of the plant’s defence response, with the ultimate goal to improve the resistance of potato to necrotrophic pathogens.

## 2. Results

### 2.1. Sequencing and Transcriptome Assembly Statistics of Potato cv. Arkula

To assess the response of potato to the pathogen *R. solani* AG3-PT, sprouts from controls (−Rs) and sprouts infected with *R. solani* AG3-PT isolate Ben3 (+Rs) were sampled at 3 and 8 dpi (*n* = 3 per sample group). Subsequently, RNA-seq was used to obtain transcriptomes of the samples. In total, 1.7 × 10^9^ reads were obtained in 36 libraries, with an average of 8.2 × 10^7^ reads per library. For one of the technical replicates from the inoculated samples at 8 dpi, only 3.6 × 10^6^ were obtained, and this library was omitted from subsequent analyses.

To optimize the existing *S. tuberosum* genome model (PGSC v4.03) for cv. Arkula, a specific reference guided genome model prediction was calculated based on the obtained transcript sequencing reads. From the hereby predicted transcripts, 68,078 contained open reading frames. Subsequently redundant transcripts were removed by means of sequence homology clustering, thereby obtaining the final list of 24,837 putative non-redundant protein coding transcripts. This non-redundant set was tested for completeness and duplications against the PGSC v4.03 reference with BUSCO (Benchmarking Universal Single-Copy Orthologs). Completeness of the model was comparable between this gene model for cv. Arkula and the PGSC v4.03 reference at 93.7% versus 87.4%, with slightly better predicted duplication rate for the cv. Arkula model with 14.5% versus 28.4%. Predicted annotations based on the GenDBE platform, including gene name, gene product, EC numbers and GO numbers if available, for the redundant and non-redundant datasets are included in Appendix A.

### 2.2. Overall Expressed Transcripts in Potato cv. Arkula

Reads per kilobase per million (RPKM) calculations were used to determine the highest expressed genes. Overall highest expressed transcripts encode abscisic stress-ripening protein 1 (ASR1) and a proteinase inhibitor 1 (Table 1). As these transcripts were highly abundant independent of inoculation with *R. solani* AG3-PT, with no significant differentially expression, they are expected to play no significantly measurable role in this host–pathogen interaction within the experimental setup and will not be further discussed. A complete list of RPKM values for the different treatments can be found in Appendix A.

Results of principal component analysis (PCA) based on the RPKM values showed a clear separation of samples by sampling time (plant developmental stage) and infection with *R. solani* AG3-PT (Figure 1). As was expected, both developmental stage and interaction with *R. solani* AG3-PT modulate the potato sprout transcriptome. Here variation of sprout developmental stage is mostly represented in principle component 1 (PC1), whereas the majority of variation between inoculated and control samples can be found within principal component 2 (PC2) (Figure 1).

### 2.3. Differentially Expressed Potato Genes during Interaction with R. solani AG3-PT

To investigate the differences in expression of genes between samples without and with *R. solani* AG3-PT inoculation, differential expression analysis was performed with DESeq. In total, 1640 differentially expressed genes (DEGs) were identified at least on pairwise comparison by DESeq analysis of the pathogen inoculated samples against the control samples at the two time points (Figure 2). During the interaction of potato sprout with *R. solani* AG3-PT, 429 and 424 genes were differentially up regulated at 3 and 8 dpi, respectively, while 324 genes were significantly up regulated at both sampling times. Contrastingly, 370 and 72 genes were differentially down regulated at 3 and 8 dpi, respectively, while 21 genes were significantly down regulated at both time points. Transcripts with the highest absolute fold changes of the pairwise comparisons have been listed in Table 2 and Table 3. Furthermore, the complete lists of DEGs for 3 and 8 dpi are listed in Appendix A.

At both time points (3 and 8 dpi), transcripts of the genes *DES*, *CYP71D7*, *CBP* and *ERF098* were differentially up regulated in pathogen-infected potato tissue (Table 2 and Table 3), whereas a transcript potentially related to the chlorophyll biosynthesis pathway (*ELIP1*) was down regulated. Other transcripts, such as of the genes *CYP71D55* (at 3 dpi) or *TPS31* (at 8 dpi), were found differentially higher expressed in infected tissue depending on the time point after pathogen inoculation. In general, many of the highly significant differentially up regulated genes are involved in the production of anti-fungal proteins or secondary metabolites with antifungal properties (e.g., *DES*, *CYP71D7*).

### 2.4. Biological Functions of DEGs

To place the transcriptional changes during the interaction of potato with *R. solani* AG3-PT into biological context, two approaches for functional enrichment of the DEGs were used. The first gene functional enrichment analysis is based on gene ontology (GO) terms and was calculated with GOrilla [11]. In a second enrichment approach, the MapMan [12] was used. Table 4 and Table 5 summarise the enriched GO terms with the lowest *p*-values for up and down regulated transcripts among the comparisons for 3 and 8 dpi. A list of all significantly enriched GO terms is available in Appendix A.

At 3 and 8 dpi, the GO terms summarizing defence response or response to stresses (GO:0006952, GO:0050896, GO:0006950) were highly significantly enriched, with GO terms of response to the biotic stimulus of fungi significantly up regulated (GO:0009607, enrichment 1.77 at 3 dpi and 2.40 at 8 dpi; GO:0043207, 1.77 (3 dpi) 2.40 (8 dpi); GO:0051707, 2.09 (3 dpi) 2.44 (8 dpi); GO:0009620, 2.27 (3 dpi) 3.10 (8 dpi)). Highly significant up regulation and a very high enrichment was also found for the GO term response to chitin (GO:0010200, 11.27 (3 dpi) 10.59 (8 dpi)) at both time points. In addition, there was also significant enrichment of the term signal transduction (GO:0007165, 1.35 (3 dpi) 1.56 (8 dpi)) with transmembrane receptor protein kinase signalling (GO:0007167, 2.11 (3 dpi) 2.49 (8 dpi); GO:0007169, 2.13 (3 dpi) 2.48 (8 dpi)), protein phosphorylation (GO:0016310, 1.48 (3 dpi) 1.73 (8 dpi); GO:0006468, 1.60 (3 dpi) 2.02 (8 dpi)) and systemic acquired resistance (GO:0009627, 5.47 (3 dpi) 3.89 (8 dpi)). Especially at 3 dpi, the regulation of hormone levels (GO:0010817, 2.61) with the biosynthesis of ethylene (ET) (GO:0009693, 16,82) and, in particular, the biosynthesis of jasmonic acid (JA) (GO:0009695, 205.04) was up regulated and more than 200-fold enriched. While at 8 dpi the regulation of hormone levels was also significant but less enriched (GO:0010817, 2.61 (3 dpi) 1.70 (8 dpi)), the GO terms for response to hormones with focus on response to salicylic acid (SA) (GO:0009751, 3.44) and response to JA (GO:0009753, 2.90) were significantly up regulated. In contrast to the up regulation of responses to the biotic stimulus, the GO terms of response to abiotic stimulus (GO:0009628, 1.43 (3 dpi) 1.57 (8 dpi); GO:0009314, 1.72 (3 dpi) 2.22 (8 dpi); GO:0009416, 1.78 (3 dpi) 2.32 (8 dpi)) were significantly down regulated at both time points tested. In addition, the enrichment of GO terms for generation of precursor metabolites and energy (GO:0006091, 3.00 (3 dpi) n.s. (8 dpi); GO:0055114, 1.81 (3 dpi) 1.86 (8 dpi); GO:0022900, 3.30 (3 dpi) n.s. (8 dpi); GO:0009767, 5.11 (3 dpi) 2.95 (8 dpi)) indicated a down regulation of photosynthetic electron transport induced by the pathogen at 3 dpi and to a lesser extent at 8 dpi. At 3 dpi, also the GO term phenylpropanoid metabolic process (GO:0009698, 4.75) and its child term phenylpropanoid biosynthesis process (GO:0009699, 5.11) were significantly reduced.

While the up regulation and highly significant enrichment of GO terms of response to biotic stimulus is similar at both time points between non-inoculated control and with *R. solani* AG3-PT inoculated samples, a strong difference was found in the enrichment of GO terms showing down regulation at 3 and 8 dpi. At 3 dpi the GO terms of response to the abiotic stimulus light (GO:0009628, GO:0009314, GO:0009416) were significantly reduced, while at 8 dpi down regulation of the response to oxidative stress (GO:0006979, GO:0042542) and response to high temperature stress (GO:0009266, GO:0009408) are most significantly enriched. In addition, at 8 dpi cellular amide metabolic processes (GO:0043603, GO:0043604, GO:0006518, GO:0043043, GO:0006412) were the most significantly enriched GO terms that were reduced.

Within a second approach, enrichment was calculated for MapMan functional bins and these were tested for significance with a Wilcoxon ranked sum test. Between control (−Rs) and *R. solani* AG3-PT-inoculated treatment (+Rs) at both time points (3 and 8 dpi), the first five functional bins with highest significance were the same and are related to stress response (Table 6 and Table 7). With regard to the biotic stress response and signalling functions, several members of large sub-families of plant receptor-like protein kinases, with putative functions in stress signal perception and transduction, were induced at both time points. Furthermore, at 3 dpi, changes within photosynthesis-related bins are more prominent as compared to time point at 8 dpi. In contrast, at 8 dpi, functional bins of protein synthesis and modification play a more significant role. The highly significant enrichment of functional bins with relation to stress response at both time points is in accordance with the GO term enrichment calculated with GOrilla. Additionally, the enrichment and down regulation of processes related to photosynthesis are in agreement in both data mining approaches. The complete list of the Wilcoxon ranked MapMan bins is included in Appendix A.

Because of the role of JA and ET in defence response to necrotrophic pathogens, GO terms related to these plant hormones were especially examined (response to ET GO:0009723; response to JA GO:0009753; Figure 3). Previous studies already highlighted the putative role of SA in disease development caused by *R. solani* AG3-PT [7] and enrichment of the GO term related to SA (response to SA GO:0009751) was found at 8 dpi; this GO term was also considered. Several clusters of genes with transcription profiles depicting co-expression of genes related in response to these plant hormones were found. Approximately one fifth of the genes share all three GO terms and are in common in the heatmaps of Figure 3, while their co-expression partners are varying, depending on their association with the GO term in question. For example, three genes homologous to *MYB44* cluster with genes homologous to serine/threonine protein kinase CTR1 and EDR1 in the response to ET heatmap, while the same three *MYB44* homologs cluster with E3 ubiquitin ligase complex component homologs in the response to JA heatmap (Figure 3A,B, pink cluster with asterisk). In the response to SA heatmap (Figure 3C), one of these *MYB44* is strongly correlated with homologs to disease resistance protein RPP8 and cysteine rich receptor-like protein kinases. This cluster analysis shows that SA- and ET/JA-mediated defence responses may act both synergistically and antagonistically [13]. A list of all transcripts included in the heatmaps (Figure 3) is available in Appendix A.

Finally, in the response of potato to the pathogen *R. solani* AG3-PT, it was found that down-regulated genes within this interaction were highly enriched with genes encoding heat shock proteins (HSPs). Figure 4 depicts the co-expression of all *HSP* candidates that were found, based on the GenDBE annotation of the potato cv. Arkula gene models. Most prominent in this analysis was a cluster of 25 co-expressed transcripts with reduced abundances at both time points after inoculation, with candidates coding for small heat shock proteins (sHSPs), HSP70, or HSP90.

## 3. Discussion

The objective of this study was to get insights into the response of potato during interaction with the necrotrophic phytopathogen *R. solani* AG3-PT. For the first time, the response of potato was investigated by analysing transcriptomes of sprouts at two time points, three and eight days post inoculation (dpi) with the pathogen. To our knowledge, this is the first RNA-seq-based comprehensive transcriptome study analysing the response of potato to *R. solani* AG3-PT.

In the field, emerging potato sprouts are colonized by *R. solani* AG3-PT, either initially from tuber-borne or soil-borne inoculum after planting. In this study, sterile grown potato tubers were sprouted and inoculated with *R. solani* AG3-PT to simulate a tuber-borne inoculum and investigate the host response. A pronounced difference was found between the transcripts of the control treatments and the treatments confronted with the pathogen. Data mining of differentially expressed genes using gene ontology enrichment analyses with GOrilla and MapMan revealed significant changes in the potato´s molecular response to *R. solani* AG3-PT at both time points (3 and 8 dpi). Here we discuss several of the transcript candidates and their potential functions related to the compatible interaction with AG3-PT.

### 3.1. Biotic Stress Response Signalling

Biotic stress response depends on signalling via plant receptor-like protein kinases (RLKs) that play vital roles in sensing outside signals with putative functions in stress signal perception and transduction [14,15]. Numerous members of such large sub-families of RLK were induced at both time points of the interaction. Noteworthy are transcripts putatively encoding for receptor-like protein 12 (RLP12), that has been identified as *CLAVATA2* (*CLV2*) in *A. thaliana* [16]. *CLV2* is implicated in distinct biological processes including plant growth and development as well as innate immunity to microbe infections [17]. However, the exact function of these potato RLP12 in signal perception and transduction has yet to be determined and will be the focus of further studies.

Chitin is a major fungal pathogen-associated molecular pattern (PAMP) that triggers PTI (PAMP triggered immunity) defence responses in plants [18]. In potato, global transcriptome analyses reveal comparable molecular signatures in the early response to either the oomycete *Phytophthora infestans*, the bacterium *Ralstonia solanacearum* or Potato virus Y infection. In addition, a weighted gene co-expression network analysis identified, amongst others, chitinases to regulate plant immune responses [19]. Several transcripts whose encoded proteins are involved in chitin signal perception and transduction were found to be induced. For example, transcripts encoding chitin elicitor receptor kinase 1 were significantly induced at both time points. While only at the later time point, LysM domain receptor-like kinase 4 homologues were up regulated, which presumably lead to the activation of plant innate immunity [20].

Further regulators of plant defence responses were found to be significantly induced: EDS1L, a positive regulator of basal resistance especially in early plant defences that also triggers hypersensitive response [21], is induced at both time points. Only at 8 dpi, RPM1-interacting protein 4 (RIN4), an essential regulator of plant defence [22], and its interacting protein NDR1 [23] are increasingly expressed. Whether these factors interact in our pathosystem, and if they play roles in resistance in case of infection of potato with *R. solani* AG3-PT, has to be experimentally determined in the future.

Another interesting induction coinciding with *R. solani* AG3-PT interaction was the strong and significant up regulation of transcripts coding for wound-induced protein 2 (WIN2). WIN proteins are a small family among the Solanaceae with high homology to chitin-binding proteins [24]. WIN2 includes a chitin binding pfam motif and partakes in the recognition of fungi-induced wounding. It is; therefore, to be assumed that WIN2 is involved in the detection of *R. solani*-induced wounding.

The up regulation of this plethora of factors involved in biotic stress response signalling indicates an induction of certain defence pathways at both time points analysed. However, a recent analysis of significantly changed expression level of rice genes in response to *R. solani* AG1 IA also revealed negative effects of the transcription factor genes OsWRKY53 and OsAKT1 on the defence response [25]. This supports the necessity that involvement of DEGs in a specific defence reply has yet to be determined, either with mutant plants altered in specific signalling components or with the analysis of potato varieties, showing a different degree of quantitative response when challenged by *R. solani*.

### 3.2. Increase of Antifungal Properties

A general strategy for defending fungal attacks is the formation of metabolites with antifungal properties. In this line of defence, the formation of reactive oxygen species (ROS) is an early event to combat the invader. Indication of such anti-fungal reactions have already been reported in *Zoysia japonica* roots 12–45 h after inoculation with *R. solani* AG1 IA, with the induction of unigenes involved in phytoalexin synthesis and oxidative burst [26]. Our results showed significantly higher expressed respiratory burst oxidase homolog (RBOH) genes especially at the early time points analysed. These genes encode NADPH oxidases, the key producers of ROS in plants, and vital performers in plant growth and signalling [27]. It has been demonstrated that especially the RBOH protein B is involved in the massive oxidative burst induced by pathogen infection in potato tuber discs [28]. However, the here analysed pathosystem led to successful attack with *R. solani* AG3-PT and a contribution of the induced expression of the RBOHs could not hinder infestation.

Concomitant with induction of oxidative burst to combat invaders is the attenuation of oxidative stress for the plant. The significantly increased expression of specific glutathione S-transferases (*GSTs*) especially at 3 dpi resembles a good example of such a strategy. *GSTs* are known as being markedly induced in the early phase of bacterial, fungal and viral infections. Their functions are linked with the decontamination of toxic substances by their conjugation with glutathione and with the attenuation of oxidative stress [29]. Furthermore, antioxidant genes of plants, including *GSTs*, were distinctly regulated during disease development in different *R. solani* pathosystems [30]. However, in *A. thaliana* the response of the promotor of *GSTF8* was used to analyse induction in infected roots, but the response differed markedly between *R. solani* strains and was not related to strain aggressiveness [31]. Therefore, a specific study of potato *GSTs* is desirable in the future in order to determine their exact contribution to reduce oxidative stress in *R. solani* interaction.

Some other genes were specifically up regulated only at the later time point 8 dpi. Amongst others are Miraculin and miraculin-like proteins possessing sequence similarities to Kunitz type proteinase inhibitors (KTI) that have been related to multiple host pathogen interactions [32]. For example *NtKTI1* from *Nicotiana tabacum*, a KTI with homology to miraculin protein revealed strong antifungal activity against *R. solani* AG-4 [33]. We therefore, expect the miraculin-like protein transcripts expressed within our experiment to play a role in the interaction of potato sprouts with *R. solani* AG3-PT.

As already mentioned, a majority of highly significant differentially higher expressed genes is presumably involved in the production of anti-fungal proteins or secondary metabolites with antifungal properties. For example, at both time points analysed the predicted gene *DES* coding for 9-divinyl ether synthase is highly and significant differentially higher expressed. This cytochrome P450 family member protein of the CYP74D subfamily was identified and characterised from elicitor-treated potato suspension cells and is involved in the synthesis of colneleic acid, a lipoxygenase-derived oxylipin implicated in functioning as a plant antimicrobial compound [34]. The cytochrome P450 superfamily (CYP) is the largest enzymatic protein family in plants, synthesising numerous secondary metabolites that function as growth and developmental signals or protect plants from various biotic and abiotic stresses [35]. Genes that, amongst others, encode CYP proteins of the CYP71D and CYP736A subfamilies were induced at both time points. With their numerous functions in biotic stress, the CYPs have an enormous potential to be used as candidates for engineering more resilient crop species in the future [36].

The up regulation of genes putatively involved in the formation of metabolites and proteins with antifungal properties demonstrates the biotic stress response and the induction of defence strategies at both time points analysed. However, the specific function of the respective gene candidates has yet to be determined (e.g., with knock-out mutant plants in interaction with *R. solani* AG3).

### 3.3. Responses Associated with Ethylene, Salicylic Acid and Jasmonic Acid

GO term enrichment analyses showed induction of pathways involved in hormonal regulation, namely ethylene (ET) and jasmonic acid (JA) biosynthetic process at 3 dpi, and JA response at 8 dpi. As was expected, these hormonal responses play a major role in the potato and *R. solani* interaction. In the past, (successful) defence against necrotrophic fungi in plants has always been dedicated to the JA and/or ET response pathways [37]. Interestingly, comparison of the transcriptomes of a moderately resistant and a susceptible cultivar of rice to *R. solani* AG1 IA revealed that the main difference between both cultivars was the timing of the response. From these results it has been suggested that the biosynthesis of JA and the synthesis of phenylalanine compounds may be important for disease tolerance [38]. A comparative transcriptome profiling of potato in response towards *P. infestans* isolates with different virulence profiles also reveals involvement of JA-, abscisic acid-, and salicylic acid (SA)-mediated signalling pathways in the response against both isolates exerting either compatible or incompatible interaction, while ethylene-mediated defence pathways were suppressed [39]. In addition, in detached lettuce leaves, inoculation with *R. solani* AG1 IB negatively regulated transcripts that putatively encode several essential proteins involved in maintaining JA homeostasis, marking an inadequate defence response [40]. More detailed studies remain to be conducted further to elucidate the roles of and the cross-talks between components of each of the pathways. More recent studies though suggest that also the SA-mediated defence plays a role of importance [7] and might even induce a certain level of resistance to necrotrophic pathogens [41]. In the here reported experiments, GO term enrichment analyses showed induction of SA-metabolic processes and response to SA at 8 dpi. The SA-mediated defence response plays a role in local and systemic-acquired resistance against biotrophic pathogens, while JA- and SA-dependent pathways can also act antagonistically [13,42]. The concomitant association of *R. solani* AG3-PT interaction with SA-mediated responses verifies that this hormone signalling pathway is involved in disease development with the soil-borne necrotrophic pathogen, indicating that crosstalk between the various plant defence pathways is even more complex. This underlines a potential systemic response of potato to *R. solani* AG3-PT attack at early phases of interaction, as also reported by Lehtonen and co-workers [8] and Genzel and co-workers [7]. Therefore, the role of the SA-dependent signalling and disease development is to be clarified in future experiments using comparative analyses of cultivars with different degree in resistance to black scurf disease.

### 3.4. Involvement of Heat Shock Proteins in the Interaction of Potato with R. solani

The potato’s response to the pathogen *R. solani* AG3-PT revealed a highly significant enrichment of down regulated genes encoding heat shock proteins (HSPs). The main function of HSPs is to act as chaperons to limit misfolding of proteins or to resolve aggregates, with the underlying molecular mechanisms being extensively reviewed [43,44]. Further inspection revealed a cluster of 25 co-expressed transcripts with reduced abundances at both time points after inoculation, with candidates coding for small heat shock proteins (sHSPs), HSP70 and HSP90 (HSP83). These HSPs are known to be involved in the response to biotic as well as abiotic stresses and are connected with plant hormone pathways. For example, HSP70 and HSP90 have been shown to be essential for the *Nicotiana benthamiana* hypersensitive response (HR) in defence against *Phytophthora infestans* and *Pseudomonas cichorii* infection [45,46]. *Pseudomonas syringae* on the other hand modifies the plant stress chaperone machinery through binding of HSP70 with the virulence effector HopI1 [47]. Infection of *A. thaliana* with *P. syringae* induces a reduced abundance of various sHSP and HSP proteins, including HSP70 [48]. Consequently, one might speculate that a yet unknown virulence effector from *R. solani* is able to bind HSPs in a HopI1-like manner, hereby leaving the binding site for recognition of transcriptional regulation freely exposed and subsequently derailing the chaperone machinery. Furthermore, it was shown that a HSP90 protein complex is necessary for the maturation and transport of a chitin recognizing PAMP in rice [49]. Preventing PAMPs from maturing would assist *R. solani* in hiding its presence. In addition, during interaction of *P. infestans* with potato, sHSP is involved in defence signalling. Here, sHSP17.8 interacts with the heat shock element domain in the StWRKY1 promoter region and helps to induce hypersensitive responses and induction of hydroxycinnamic acid amides and defence by secondary cell wall thickening [50]. Interestingly, sHSP17.8 is only induced in the resistant genotype and was completely absent in the susceptible genotype. Whether HSP or HSP-regulation in potato is truly a target for putative *R. solani* effectors cannot be validated based on transcriptome data alone. Previous studies on the interaction between *R. solani* AG1 and lettuce revealed the expression of potential effector proteins during interaction with the plant [40]. Future experiments should be focussed on the protein level to adequately test the involvement of HSPs and their regulators as potential targets.

## 4. Materials and Methods

### 4.1. Plant Treatment and Sampling

The potato cultivar ‘Arkula’ (Norika GmbH, Sanitz, Germany), which has been shown to be susceptible to *R. solani* AG3 [7,10], was used to study the plant response in compatible interaction with *R. solani* AG3-PT isolate Ben3. To ensure pathogen-free plant material, potato tubers were produced from in vitro plantlets (kindly provided by Norika GmbH). Plant cultivation and inoculation were the same as described by Zrenner and co-workers [10]. Sprouts of 10 tubers per replicate were sampled and pooled at 3 and 8 days after pathogen inoculation or control treatment (dpi). At the first sampling time point, no lesions were observed. In contrast, at the second time point, individual sprouts of inoculated samples did show lesions. These sprouts were divided into parts with and without lesion and the parts without lesion were pooled per replicate and used for RNA-seq analysis. Subsequently, sprout samples were immediately shock frozen in liquid nitrogen and stored at −80 °C. Each sample was ground using a mixer mill (2 min, 30 × s^−1^; Retsch MM400, Haan, Germany) with two grinding balls (7, 3 mm; Askubal, Korntal-Münchingen, Germany) under constant cooling in liquid nitrogen.

### 4.2. RNA Extraction

Total RNA was extracted from 70–90 mg of ground sprout material (without lesions) using the RNeasy Plant Mini Kit (Qiagen, Hilden, Germany) including DNase treatment (Qiagen). Quantity and quality of the RNA was checked with the bioanalyzer (Agilent Technologies Deutschland GmbH, Waldbronn, Germany) and samples were stored at −80 °C until use.

### 4.3. Sequencing of cDNA Libraries

cDNA library preparation and sequencing were performed at the IIT GmbH (Bielefeld, Germany). For cDNA library synthesis, total RNA was used with the TruSeq^®^ mRNA Sample Preparation Kit (stranded) (Illumina Inc., San Diego, CA, USA). From each biological replicate, 3 technical replicates were generated resulting in a total of 36 libraries. The libraries were pooled in three sets of 12 and each pool was sequenced twice on the Illumina HiSeq 1500 platform (Illumina). The cDNA libraries were single-end sequenced in rapid mode with 1 × 50 cycles. Base calling and de-multiplexing were done with in-house software based on CASAVA 1.8.2. (Illumina).

### 4.4. Mapping and Gene Prediction

In order to optimize the downstream transcriptome analysis, a reference-guided de novo genome model was predicted using the potato reference genome chromosomal pseudomolecules of doubled monoploid *S. tuberosum* Group Phureja clone DM1-3 (DM) (potato genome sequencing consortium v.4.03, PGSC v4.03) as template [51,52]. The sequencing reads, after quality filtering with Trimmomatic, were mapped with Tophat2 onto the PGSC v4.03 reference [53]. Subsequently Cufflinks (v2.1.1) [54] in de novo mode was used for transcript prediction. These steps were performed on each sequencing file separately, to improve the detection of under expressed transcripts. The various Cufflinks predictions were merged with Cuffmerge (v1.0.0). Open reading frames were selected from the predicted transcripts with Transdecoder (v2.0.1) [55], the resulting gff3 file was imported in the GenDBE platform [56] for annotation predictions. For further transcriptional analysis, representative sequences were selected by means of CD-hit sequence homology clustering (homology cut off 0.8) [57]. The non-redundant transcriptome derived thereof was tested for completeness and redundancies, and compared to the reference PGSC v4.03 transcriptome with the BUSCO (v3.0.0) application [58].

### 4.5. Expression Analysis and Functional Enrichment

To calculate transcription levels and differential transcription, RPKM (reads per kilobase per million) values and DESeq calculations were used, as implemented within the ReadXplorer platform (v2.2) [59,60,61]. For DESeq, genes with a false discovery rate (FDR) adjusted *p*-value of less than 0.05 [62] were deemed differentially expressed. Principle component analysis (PCA) based on RPKM values were generated with the ClustVis Docker package [63,64] to test the distribution between the replicates.

Gene functional annotation of differential expressed transcripts were based on both gene ontology (GO) and MapMan. GOrilla was used for functional enrichment of GO terms [11]. The translated protein sequences from the reduced list of potato transcripts were blasted against the *Arabidopsis thaliana* proteins extracted from the UniProtKB database (www.uniprot.org; accessed on 5 February 2016) to retrieve homologues candidates (maximum e-value threshold 10^−11^). These candidates were sorted in order of the DESeq output and fed into the GOrilla web platform, for GO enrichment. For pathway analysis with MapMan, first the homology clustered protein list was annotated with the Mercator pipeline [65]. The resulting mapping file was imported into the MapMan [12] platform together with the DESeq derived fold changes. All non-redundant transcripts were used in the Wilcoxon rank sum test. Bins with a Benjamini Hochberg corrected *p*-value of 0.05 or less were deemed significant. Heat maps were plotted with ClustVis [64].

### 4.6. Validating Expression Differences with RT-qPCR

RNA extraction and quality control was performed as described. Single-stranded cDNA synthesis of 1 µg of total RNA using iScript cDNA Synthesis Kit (Bio-Rad Laboratories GmbH, Feldkirchen, Germany) in a 25 µL reaction was done following manufacturer´s instructions and diluted 10-fold. RT-qPCR using 96-well reaction plates and Thermal Cycler CFX96 C1000 Touch (Bio-Rad) was performed with the thermal profile 95 °C for 5 min, 40 cycles of 95 °C for 15 s and 60 °C for 1 min, followed by dsDNA melting curve analysis. Each reaction of 10 µL volume contained 200 nM of each primer, 3 µL of cDNA (1:10) and 5 µL of Sensi Fast SYBR NO ROX Kit (Bioline GmbH, Luckenwalde, Germany). Data collection and analysis was done with CFX Manager Software 3.0 (Bio-Rad). Three biological replicates were measured in duplicates, uninfected control plants and also non-template controls were included.

Relative transcript levels of three candidate genes with different expression patterns were normalized on the basis of expression of the invariant control actin (*ACT*) as in [7]. Oligonucleotide primer sets for *HSP70* (StHSP70f-GCAGGAACAGTAACCACAGCG; StHSP70r-GGTAGTTCCGGGTCCTGGTG), *DES* (StDESf-TCTCTTCAGGTTTGTGGGCGA; StDESr-AATGGTACTGGTGGGCGAAG), and *CYP71D55* (StCYP71D55f-AGCACAAACAAAATCGAGCGGA; StCYP71D55r-CCGCGGAGAACATGTCGACT) were tested for reliable amplifications with efficiencies of close to 2. ΔCq was calculated as the difference between control and target products (ΔCq = Cq*_ACT_* − Cq*_gene_*). Differences in relative expression levels between the treated samples were calculated as ΔΔCq = ΔCq (+Rs) − ΔCq (−Rs) and compared with the respective DESeq2 analysis (Table 8). The calculated ΔΔCq values of the three candidate genes were in accordance with the respective log2fold change values computed with DESeq2 and reassure expression differences.

## 5. Conclusions

The results from the current experiment highlight the complexity of host-pathogen interactions and the crosstalk between the defence response pathways. At 3 dpi, potato induces general defence strategies. In contrast, at 8 dpi this response appears to be more focussed towards the SA-mediated defence. Although several recent studies imply, that SA-mediated responses might provide a certain level of resistance at the onset of infection [41], it is plausible that defence at more advanced infection stages would benefit from JA-mediated responses. Additionally it should be noted that inducing the SA response pathway without inducing the senescence pathway is a fine balancing act [66]. From our data, no clear shift towards a JA-mediated defence response could be observed at the more advanced stage of interaction. Therefore, we hypothesise that the sensitive potato cv. Arkula is incapable of timely attenuation between the two major defence strategies during progressed interaction with *R. solani* AG3-PT. Alternatively, it is possible that *R. solani* AG3-PT successfully obscures its identity or actively sabotages the potato defence by means of effector proteins. These results extend upon the ever growing number of experiments dedicated to *R. solani* related pathosystem and ultimately leads to a better understanding of this phytopathogen species complex and possible control strategies.

## Figures and Tables

**Figure 1 ijms-22-03094-f001:**
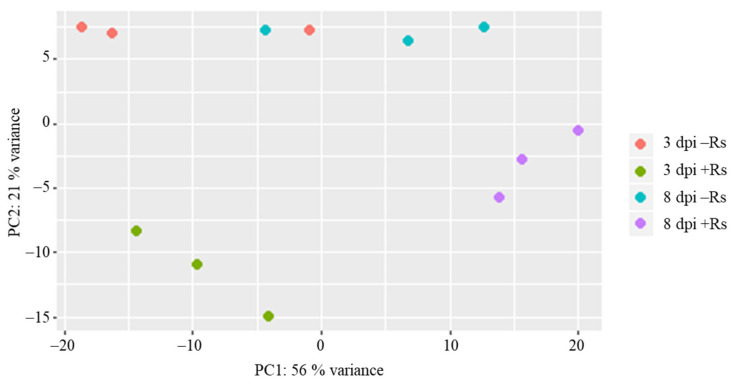
Two-dimensional principal component analysis (PCA) based on RPKM values in samples of potato sprouts (cv. Arkula) without (−Rs) and with *Rhizoctonia solani* AG3-PT inoculation (+Rs) at three and eight days post pathogen inoculation (dpi).

**Figure 2 ijms-22-03094-f002:**
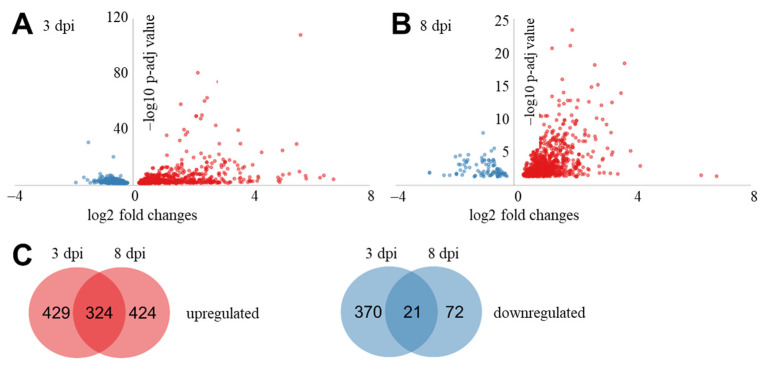
Differentially expressed genes (DEGs) of potato sprouts (cv. Arkula) in response to *Rhizoctonia solani* AG3-PT at 3 and 8 dpi. (**A,B**) DESeq-based volcano plot of DEGs between control and with *R. solani* inoculated samples. (**C**) Venn diagram of DEGs between 3 and 8 dpi. Up regulated DEGs in red and down regulated DEGs in blue.

**Figure 3 ijms-22-03094-f003:**
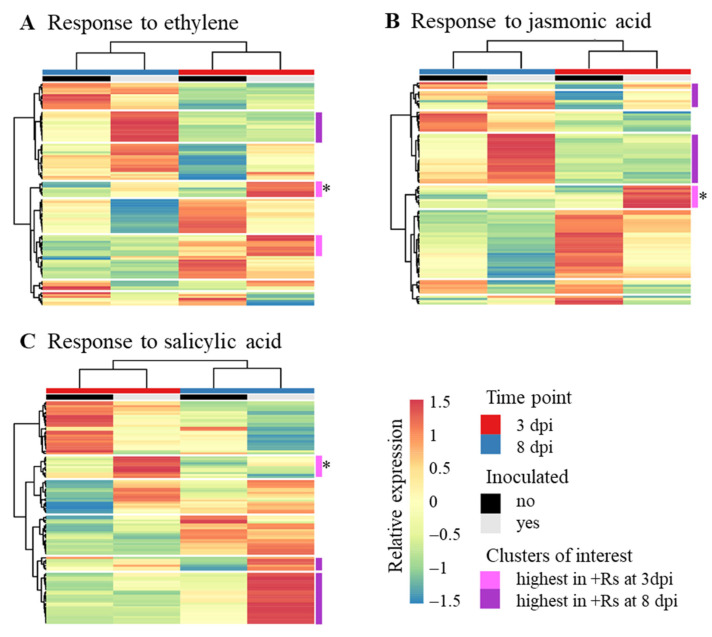
Heatmaps of relative expression levels of genes of particular gene ontology (GO) terms. Shown are relative expression levels of genes in control (−Rs) and in with *Rhizoctonia solani* AG3-PT (+Rs)-inoculated potato sprouts (cv. Arkula) at 3 and 8 dpi. (**A**) Response to ethylene GO:0009723. (**B**) Response to jasmonic acid GO:0009753. (**C**) Response to salicylic acid GO:0009751. Relative expression levels are based on reads per kilobase per million (RPKM) values. Both rows and columns are clustered by Euclidean distance and Ward linkage. Clusters of interest are highlighted in pink and lilac, * asterisks mark clusters of specific interest. A listing of the included transcripts can be found in Appendix A.

**Figure 4 ijms-22-03094-f004:**
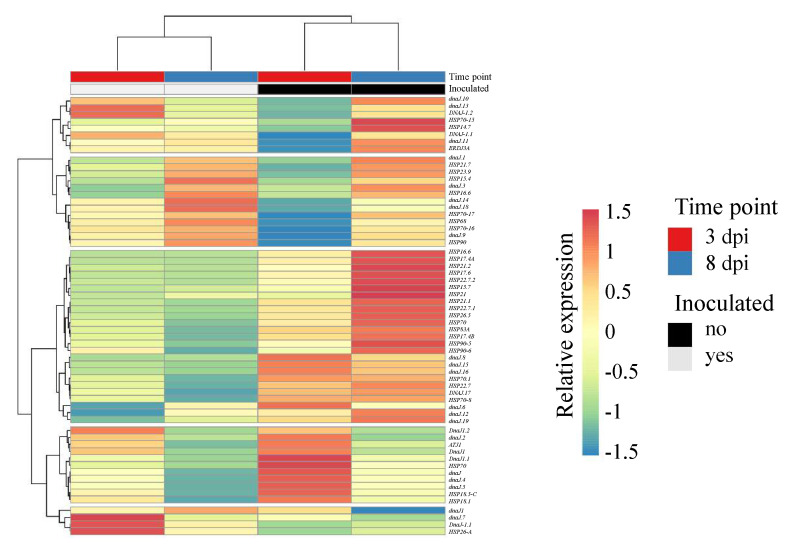
Heatmap clustering of heat shock protein genes comparing control (−Rs) and with *Rhizoctonia solani* AG3-PT (+Rs)-inoculated potato sprouts (cv. Arkula) at 3 and 8 dpi. Reads per kilobase per million (RPKM)-based clustering of both rows and columns are performed by Euclidean distance and Ward linkage. A listing of the included transcripts can be found in Appendix A.

**Table 1 ijms-22-03094-t001:** Overall highest expressed potato (cv. Arkula) transcripts in treatments without (−Rs) and with *Rhizoctonia solani* AG3-PT (+Rs) inoculation. Averages of (−Rs) and (+Rs) at three days post inoculation (dpi) and 8 dpi based on reads per kilobase per million (RPKM) levels.

Feature	Gene Product	−Rs	+Rs	Average
TCONS_00022201_m.32854	Abscisic stress-ripening protein 1	6687	5280	5983
TCONS_00049449_m.67948	Non-specific lipid-transfer protein 2	7349	3723	5536
TCONS_00046088_m.64962	Proteinase inhibitor 1	3979	3740	3860
TCONS_00022130_m.32684	Histidine-rich glycoprotein	2305	3558	2931
TCONS_00058960_m.82079	Catalase isozyme 2	2607	3138	2872
TCONS_00046852_m.60760	Metallothionein-like protein type 2 B	3083	2618	2851
TCONS_00046087_m.64961	Proteinase inhibitor 1	3029	2596	2813
TCONS_00046864_m.60797	Heat shock cognate 70 kDa protein 2	2546	2401	2474
TCONS_00002891_m.7661	Translationally-controlled tumour protein	2356	2144	2250
TCONS_00039696_m.54216	Heat shock cognate protein 80	1988	1723	1855

**Table 2 ijms-22-03094-t002:** Transcripts with highest absolute fold changes between control (−Rs) and with *Rhizoctonia solani* AG3-PT (+Rs)-inoculated potato samples at 3 dpi according to DESeq.

Loci	Base Mean	Log2 Fold Change	*p*-Adjusted	Gene Name (Predicted)	Gene Product (Predicted)
TCONS_00004194_m.10757	563	6.8	1.0 × 10^−4^	*DES*	9-Divinyl ether synthase
TCONS_00041095_m.57761	1959	6.4	1.1 × 10^−7^	*CYP71D7*	Cytochrome P450 71D7
TCONS_00047931_m.63689	42	6.3	4.9 × 10^−6^	*ERF098*	Ethylene-responsive transcription factor
TCONS_00040978_m.57527	1091	5.9	3.0 × 10^−5^	*AIM1*	Peroxisomal fatty acid β-oxidation protein
TCONS_00008898_m.1264	696	5.8	2.6 × 10^−7^	*CYP71D55*	Premnaspirodiene oxygenase
TCONS_00053685_m.73894	2282	5.7	6.2 × 10^−109^	*CBP*	Citrate-binding protein
TCONS_00032279_m.47383	97	5.6	1.9 × 10^−10^	*PUB24*	E3 ubiquitin-protein ligase PUB24
TCONS_00038986_m.52164	437	5.5	2.9 × 10^−30^	*GDS*	(-)-Germacrene D synthase
TCONS_00048229_m.64462	371	5.3	8.8 × 10^−9^	*HIDM*	2-Hydroxyisoflavanone dehydratase
TCONS_00021788_m.31824	2715	5.1	4.3 × 10^−8^	*CYP71D7*	Cytochrome P450 71D7
TCONS_00038015_m.49462	3156	−1.2	8.7 × 10^−3^	*At4g36180*	Probable LRR receptor-like ser/thr-PK
TCONS_00010359_m.16650	723	−1.3	2.4 × 10^−3^	*STR15*	Rhodanese-like domain protein 15; chloro
TCONS_00047559_m.62592	403	−1.3	5.0 × 10^−4^	*HSP17.8*	17.8 kDa class I heat shock protein
TCONS_00058679_m.81355	148	−1.3	3.0 × 10^−2^	*DMG400005010*	Uncharacterized protein
TCONS_00048318_m.64746	19,682	−1.3	1.9 × 10^−3^	*ELIP1*	Early light-induced protein; chloroplastic
TCONS_00037799_m.48946	221	−1.5	2.7 × 10^−2^	*NCS1*	S-norcoclaurine synthase 1
TCONS_00008336_m.11190	2494	−1.5	2.3 × 10^−31^	*SN2*	Snakin-2
TCONS_00019937_m.26919	658	−1.5	1.9 × 10^−4^	*DMG400014234*	Uncharacterized protein
TCONS_00002388_m.6328	1515	−1.7	2.7 × 10^−6^	*WSD1*	O-acyltransferase WSD1
TCONS_00035423_m.48440	244	−1.9	1.6 × 10^−2^	*At5g33370*	GDSL esterase/lipase

**Table 3 ijms-22-03094-t003:** Transcripts with highest absolute fold changes between control (−Rs) and with *Rhizoctonia solani* AG3-PT (+Rs)-inoculated potato samples at 8 dpi according to DESeq.

Loci	Base Mean	Log2 Fold Change	*p*-Adjusted	Gene Name (Predicted)	Gene Product (Predicted)
TCONS_00021878_m.32075	22	6.8	4. × 10^−2^	*TPS31*	Viridiflorene synthase
TCONS_00016163_m.24776	16	6.3	3.3 × 10^−2^	*MCL*	Miraculin
TCONS_00047931_m.63689	22	4.3	1.3 × 10^−3^	*ERF098*	Ethylene-responsive transcription factor
TCONS_00004194_m.10757	126	3.9	6.8 × 10^−6^	*DES*	9-Divinyl ether synthase
TCONS_00053685_m.73894	735	3.7	5.1 × 10^−19^	*CBP*	Citrate-binding protein
TCONS_00009061_m.13072	1210	3.6	1.5 × 10^−14^	*PLP1*	Patatin-like protein 1
TCONS_00054756_m.76933	52	3.5	2.0 × 10^−2^	*Kwl1*	Kiwellin
TCONS_00050984_m.71786	119	3.3	3.6 × 10^−13^	*CYP98A2*	Cytochrome P450 98A2
TCONS_00018583_m.23294	490	3.3	2.7 × 10^−5^	*DREB1A*	Dehydration-response element-binding 1A
TCONS_00041095_m.57761	207	3.3	1.2 × 10^−8^	*CYP71D7*	Cytochrome P450 71D7
TCONS_00050284_m.70004	31	3.3	1.7 × 10^−2^	*UGT73D1*	UDP-glycosyltransferase 73D1
TCONS_00059966_m.79472	425	−1.8	3.1 × 10^−5^	*BBX22*	B-box zinc finger protein 22
TCONS_00019878_m.26786	9475	−1.9	7.6 × 10^−8^	*HSP70*	Heat shock cognate 70 kDa protein
TCONS_00025911_m.35167	84	−1.9	2.2 × 10^−2^	*PER64*	Peroxidase 64
TCONS_00047789_m.63274	364	−1.9	1.8 × 10^−2^	*At4g02900*	CSC1-like protein
TCONS_00047789_m.63275	364	−1.9	1.8 × 10^−2^	*-*	Hypothetical protein
TCONS_00048318_m.64746	15,884	−2.0	5.0 × 10^−5^	*ELIP1*	Early light-induced protein; chloroplastic
TCONS_00059918_m.79395	402	−2.2	3.3 × 10^−2^	*PER27*	Peroxidase 27
TCONS_00053904_m.74476	6317	−2.4	3.2 × 10^−2^	*HSP22.7*	22.7 kDa class IV heat shock protein
TCONS_00059319_m.82977	56	−2.9	1.3 × 10^−2^	*EXT2*	Extensin-2

**Table 4 ijms-22-03094-t004:** Most significantly affected gene ontology (GO) terms among the up and down regulated potato transcripts between control (−Rs) and with *Rhizoctonia solani* AG3-PT (+Rs) inoculated samples at 3 dpi.

Up/Down	GO Term	GO Term Name	FDR ^1^ q-Value	Enrichment
+	GO:0006952	defence response	1.8 × 10^−14^	1.9
+	GO:0050896	response to stimulus	3.9 × 10^−13^	1.5
+	GO:0006468	protein phosphorylation	9.5 × 10^−11^	1.6
+	GO:0043207	response to external biotic stimulus	2.0 × 10^−10^	1.8
+	GO:0009607	response to biotic stimulus	1.6 × 10^−10^	1.8
+	GO:0016310	phosphorylation	1.4 × 10^−10^	1.5
+	GO:0051707	response to other organisms	1.8 × 10^−10^	2.1
+	GO:0051704	multi-organism process	3.2 × 10^−10^	1.6
+	GO:0010200	response to chitin	3.9 × 10^−10^	11.3
+	GO:0006950	response to stress	2.9 × 10^−9^	1.5
−	GO:1901700	response to oxygen-containing compound	4.2 × 10^−8^	2.3
−	GO:0022900	electron transport chain	3.9 × 10^−8^	3.3
−	GO:0009699	phenylpropanoid biosynthetic process	1.4 × 10^−8^	5.1
−	GO:0009698	phenylpropanoid metabolic process	6.7 × 10^−9^	4.8
−	GO:0009314	response to radiation	5.9 × 10^−9^	1.7
−	GO:0009628	response to abiotic stimulus	3.8 × 10^−9^	1.4
−	GO:0009416	response to light stimulus	3.1 × 10^−10^	1.8
−	GO:0055114	oxidation-reduction process	2.3 × 10^−10^	1.8
−	GO:0006091	generation of precursor metabolites and energy	1.6 × 10^−10^	3.0
−	GO:0009767	photosynthetic electron transport chain	1.9 × 10^−12^	5.1

^1^ False discovery rate adjusted *p*-value.

**Table 5 ijms-22-03094-t005:** Most significantly affected gene ontology (GO) terms among the up and down regulated potato transcripts between control (−Rs) and with *Rhizoctonia solani* AG3-PT (+Rs)-inoculated samples at 8 dpi.

Up/Down	GO Term	GO Term Name	FDR ^1^ q-Value	Enrichment
+	GO:0006952	defence response	5.3 × 10^−40^	3.0
+	GO:0050896	response to stimulus	2.6 × 10^−23^	1.6
+	GO:0010200	response to chitin	3.6 × 10^−23^	10.6
+	GO:0006950	response to stress	2.0 × 10^−22^	1.8
+	GO:0051707	response to other organisms	1.4 × 10^−21^	2.4
+	GO:0009607	response to biotic stimulus	4.2 × 10^−21^	2.4
+	GO:0043207	response to external biotic stimulus	4.9 × 10^−21^	2.4
+	GO:0051704	multi-organism process	2.2 × 10^−20^	1.9
+	GO:0006468	protein phosphorylation	1.3 × 10^−15^	2.0
+	GO:0042493	response to drug	2.3 × 10^−15^	4.4
−	GO:0055114	oxidation-reduction process	1.7 × 10^−7^	1.9
−	GO:0006979	response to oxidative stress	1.9 × 10^−8^	9.1
−	GO:0042542	response to hydrogen peroxide	1.6 × 10^−8^	28.4
−	GO:0009266	response to temperature stimulus	9.6 × 10^−10^	7.9
−	GO:0043603	cellular amide metabolic process	5.2 × 10^−10^	1.7
−	GO:0009408	response to heat	3.4 × 10^−12^	16.1
−	GO:0006518	peptide metabolic process	1.5 × 10^−12^	2.0
−	GO:0043604	amide biosynthetic process	4.7 × 10^−13^	2.0
−	GO:0043043	peptide biosynthetic process	1.2 × 10^−15^	2.2
−	GO:0006412	translation	4.2 × 10^−16^	2.2

^1^ False discovery rate adjusted *p*-value.

**Table 6 ijms-22-03094-t006:** Most significant MapMan bins comparing control (−Rs) and with *Rhizoctonia solani* AG3-PT (+Rs)-inoculated potato sprouts (cv. Arkula) at 3 dpi.

Bin ^1^	MapMan Bin Name	Elements ^2^	*p*-Value ^3^
20.1.7	Stress.biotic.PR-proteins	365	0.0
30	Signalling	1222	0.0
20	Stress	894	0.0
20.1	Stress.biotic	543	0.0
30.2	Signalling.receptor kinases	534	0.0
1	PS	170	7.4 × 10^−54^
1.1	PS.lightreaction	113	1.2 × 10^−43^
1.1.1	PS.lightreaction.photosystem II	39	5.6 × 10^−16^
1.1.1.2	PS.lightreaction.photosystem II.PSII polypeptide subunit	30	2.8 × 10^−13^
29.2.1.1.1	Protein.synthesis.ribosomal protein.prokaryotic.chloro	51	4.4 × 10^−11^
29.2.1.1	Protein.synthesis.ribosomal protein.prokaryotic	99	3.0 × 10^−10^
30.2.11	Signalling.receptor kinases.leucine rich repeat XI	63	5.2 × 10^−10^
1.1.2	PS.lightreaction.photosystem I	22	3.1 × 10^−9^
29.2.1	Protein.synthesis.ribosomal protein	216	4.1 × 10^−9^
30.2.24	Signalling.receptor kinases.S-locus glycoprotein like	56	2.7 × 10^−8^
1.1.2.2	PS.lightreaction.photosystem I.PSI polypeptide subunits	15	2.0 × 10^−7^
30.2.20	Signalling.receptor kinases.wheat LRK10 like	25	3.0 × 10^−7^
26.9	Misc.glutathione S transferases	50	3.0 × 10^−7^
29.2	Protein.synthesis	412	3.0 × 10^−7^
26.2	Misc.UDP glucosyl and glucoronyl transferases	204	3.0 × 10^−7^

^1^ Numbers of the 20 most significant bins, according to MapMan ontology. ^2^ Number of elements in the respective bin of the MapMan ontology. ^3^ Benjamini-Hochberg corrected *p*-values of Wilcoxon ranked sum test for −Rs 3 dpi vs. +Rs 3 dpi.

**Table 7 ijms-22-03094-t007:** Most significant MapMan bins comparing control (−Rs) and with *Rhizoctonia solani* AG3-PT (+Rs)-inoculated potato sprouts (cv. Arkula) at 8 dpi.

Bin ^1^	MapMan Bin Name	Elements ^2^	*p*-Value ^3^
20.1.7	Stress.biotic.PR-proteins	366	0.0
30	Signalling	1222	0.0
20	Stress	895	0.0
20.1	Stress.biotic	544	0.0
30.2	Signalling.receptor kinases	534	0.0
29.2.1	Protein.synthesis.ribosomal protein	216	2.7 × 10^−30^
29.2	Protein.synthesis	412	1.2 × 10^−29^
1	PS	170	1.4 × 10^−22^
1.1	PS.lightreaction	113	6.5 × 10^−18^
35	Not assigned	11290	1.4 × 10^−17^
29.2.1.1	Protein.synthesis.ribosomal protein.prokaryotic	99	8.0 × 10^−16^
29.2.1.2	Protein.synthesis.ribosomal protein.eukaryotic	106	3.3 × 10^−14^
27.3.32	RNA.regulation transcription.WRKY transcription factor	62	1.6 × 10^−13^
35.1.5	Not assigned. pentatricopeptide (PPR) repeat-containing	422	1.6 × 10^−13^
30.2.24	Signalling.receptor kinases.S-locus glycoprotein like	56	2.6 × 10^−13^
20.1.2	Stress.biotic.receptors	33	1.3 × 10^−11^
30.2.20	Signalling.receptor kinases.wheat LRK10 like	25	5.4 × 10^−10^
30.2.8	Signalling.receptor kinases.leucine rich repeat VIII	36	5.1 × 10^−9^
29.4	Protein.postranslational modification	607	5.3 × 10^−9^
29.2.1.1.1	Protein.synthesis.ribosomal protein.prokaryotic.chloro	51	8.7 × 10^−9^

^1^ Numbers of the 20 most significant bins, according to MapMan ontology. ^2^ Number of elements in the respective bin of the MapMan ontology. ^3^ Benjamini-Hochberg corrected *p*-values of Wilcoxon ranked sum test for −Rs 8 dpi vs. +Rs 8 dpi.

**Table 8 ijms-22-03094-t008:** Validation of DESeq2 analysis with RT-qPCR of three genes of interest.

SeqName	∆Cq (a) ^1^	∆Cq (b) ^1^	∆∆Cq ^2^	BaseMean ^3^	Log2fold Change ^3^	Comparison (a):(b) ^4^
*TCONS_00019878_m.26787; HSP70*	6.9 ± 0.12	7.8 ± 0.56	−1.86 *	10605	−0.6	(3 dpi):(−Rs)
*TCONS_00019878_m.26787; HSP70*	6.3 ± 0.80	7.1 ± 2.11	−0.76	9475	−1.9	(8 dpi):(−Rs)
*TCONS_00004194_m.10757; DES*	4.0 ± 2.23	−0.8 ± 0.83	4.01	563	6.8	(3 dpi):(−Rs)
*TCONS_00004194_m.10757; DES*	1.4 ± 0.29	−3.5 ± 2.09	4.87 *	126	3.9	(8 dpi):(−Rs)
*TCONS_00008898_m.12640; CYP71D55*	2.4 ± 2.00	n.d.	-	1959	5.8	(3 dpi):(−Rs)
*TCONS_00008898_m.12640; CYP71D55*	0.02 ± 0.4	n.d.	-	120	3.0	(8 dpi):(−Rs)

^1^ ΔCq, relative expression value calculated as Cq*_ACT_* − Cq*_gene_*. ^2^ ΔΔCq, differences in relative expression levels calculated as ΔCq (a) − ΔCq (b). ^3^ BaseMean average and Log2fold change, calculated with DESeq2, listed in Appendix A. ^4^ Potato sprouts (cv. Arkula) from controls (−Rs) or in response to *Rhizoctonia solani* AG3-PT (+Rs) at 3 and 8 dpi. * stands for significant difference of ΔΔCq (*n* = 3; *p* ≤ 0.05; *t*-test). n.d., not detected.

## Data Availability

The sequencing raw data for all libraries presented in this study are openly available on the EBI ArrayExpress server with the accession E-MTAB-7137 at https://www.ebi.ac.uk/arrayexpress/experiments/E-MTAB-7137. The gene prediction has been made available on the CoGe platform under genome ID 52025.

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
