# Peer review of "Transcriptional Changes in Potato Sprouts upon Interaction with Rhizoctonia solani Indicate Pathogen-Induced Interference in the Defence Pathways of Potato"

_ijms, 2021, doi:10.3390/ijms22063094_

Round 1

Reviewer 1 Report

The present manuscript describes the transcriptome in potato sprouts exposed to Rhizoctonia solani. The topic is relevant and in the last decade several studies have been done on the topic. The discussion of this manuscript needs large improvements and integration with previous reports. The researchers should describe much clearly what is novel in this manuscript because several points had been described before and are not novel. The transcriptome of multiple plants following exposure to Rhizoctonia solani have been previously (e.g. see manuscripts s41598-019-43706-5, s41598-017-10804-1, 10.3389/fmicb.2016.00708, 10.1007/s10142-019-00677-0, 10.3389/fpls.2017.01422; some of these manuscripts are not listed as references and it should be added and discussed). The activity and effects of some transcriptional factors have been recently tested and confirmed experimentally in rice and the results were similar to some of the results presented here (Yuan et al 2020, 10.1007/s11816-020-00630-9). These studies are much more useful than the pure description of the transcriptome profiles as presented in this manuscript. Anyway it would be useful a good discussion of such results and similarities. The transcriptome of potato exposed to Ralstonia solanacearum, Phytophthora infestans, Helminthosporium solani or other diseases was already published (e.g. see 10.1007/s00425-020-03471-6, 10.1534/g3.119.400818, 10.1094/MPMI-03-17-0062-R, 10.1186/s12864-015-1460-1) and the results agree with the results presented in this manuscript; such results were not discussed. Other important manuscripts not included as references are s41598-019-55734-2and 10.1016/j.funbio.2014.06.007.

Additional comments

1) Table 8. RT-qPCR and DESeq2 results should agree but the agreement is it low for some cases presented in this table. Can we really trust the results with changes between -2 and 3 described above in the manuscript? Some differences may not be as relevant as stated in other tables.

2) Three replicates were used for each comparison (3 versus 8 dpi; -Rs versus +Rs). Is it possible to calculate standard deviations for the log2 fold changes presented in the tables?

Author Response

Please see my response file with our point-by-point response

Reviewer 2 Report

This manuscript has multiple flaws. An early time point of sample collection for early plant response is missing here.

Did you use a water-inoculated control to filter out the genes related to other processes? A water-inoculated control must be added to the experiment.

Better analysis and representation of the transcriptome results are required. For example, including a MapMan diagram showing up-regulated and down-regulated genes by varying color intensity.

Most importantly, including a concluding pathway map based on the findings of this study would be great.

Line# 19: 3 days (72h) is not an early phase. It would be good to see close to fungal spore germination and the start of colonization.  

Line# 71: “…host response due to pathogen……”

Line# 82: What was the basis of time point selection? Why did you select 3 dpi and 8 dpi?

Author Response

(The authors gave the same response as above.)

Round 2

Reviewer 1 Report

The manuscript has been improved and it can be published.